# Process Optimization of Via Plug Multilevel Interconnections in CMOS Logic Devices

**DOI:** 10.3390/mi11010032

**Published:** 2019-12-25

**Authors:** Yinhua Cui, Jeong Yeul Jeong, Yuan Gao, Sung Gyu Pyo

**Affiliations:** 1School of Integrative Engineering, Chung-Ang University, Seoul 06974, Korea; yinhua0822@gmail.com (Y.C.); gaoyuan4025@gmail.com (Y.G.); 2Process Development Center, Magnachip Semiconductor, Seoul 15213, Korea; trinitysg@naver.com

**Keywords:** multilevel metallization, logic device, RF etching

## Abstract

This paper reports on the optimization of the device and wiring in a via structure applied to multilevel metallization (MLM) used in CMOS logic devices. A MLM via can be applied to the Tungsten (W) plug process of the logic device by following the most optimized barrier deposition scheme of RF etching 200 Å IMP Ti (ion metal plasma titanium) 200 Å CVD TiN (titanium nitride deposited by chemical vapor deposition) 2 × 50 Å. The resistivities of the glue layer and barrier, i.e., IMP Ti and CVD TiN, were 73 and 280 μΩ·cm, respectively, and the bottom coverages were 57% and 80%, respectively, at a 3.2:1 aspect ratio (A/R). The specific resistance of the tungsten film was approximately 11.5 μΩ·cm, and it was confirmed that the via filling could be performed smoothly. RF etching and IMP Ti should be at least 200 Å each, and CVD TiN can be performed satisfactorily with the existing 2 × 50 Å process. Tungsten deposition showed no difference in the via resistance with deposition temperature and SiH_4_ reduction time. When the barrier scheme of RF etching 200 Å IMP Ti 200 ÅCVD TiN 2 × 50 Å was applied, the via resistance was less than 20 Ω, even with a side misalignment of 0.05 μm and line-end misalignment of ~0.1 μm.

## 1. Introduction

With increased integration of logic devices, the number of metal wiring layers in multilevel metallization (MLM) has increased from 5 layers to 6–12 layers, and the wiring width has become low [1,2]. In addition, the hole sizes of vias used to connect the wiring layers have decreased, whereas the thickness of the inter-metal dielectric (IMD) [3], which determines the height of the vias, has hardly changed, resulting in a sharp increase in the aspect ratio (A/R; via height/via hole size) [4,5]. In this situation, the via has reached its filling limit with the conventional Al reflow or 2-step Al deposition process [6,7]. Alternatively, the plug process using chemical vapor deposition tungsten (CVD W) has been applied after deposition of a barrier film [8]. The need has arisen. In existing logic devices, an MLM structure is formed by a 5-level via plug using a barrier film and CVD W [9,10]. The overall trend of the W plug process is common when Al is used in the metal wiring [11,12]. Therefore, W plugs are also applied to the via used in MLM structures of logic devices, and it is reasonable to apply ionized metal plasma physical vapor deposition (IMP PVD) Ti and titanium nitride deposited by chemical vapor deposition (CVD TiN) structures to logic devices [13,14].

Via filling processes include the W plug fill process, the Al plug fill process, and the Cu damascene fill process. In the W plug fill process, the CVD W process is applied in terms of filling [15,16]. It is essential to derive the optimum conditions of the via profile photo/etch integration process, the glue layer Ti, and the barrier metal TiN process to optimize the process. If the Ti/TiN process is not optimized, the Volcano effect [17,18] can occur, causing a fatal defect that can cause the device to fail. In the Al plug fill process, a two-step process of CVD Al/PVD Al is applied, and the selection of precursor, flow fill method, glue layer, and pattern profile applied to CVD Al act as process variables. Currently, the pattern of sub-0.2mm (aspect ratio) 4 vias is a voidless and stable filling process with an integrated CVD Al/PVD AlCu approach process [19,20]. The Cu damascene process is a completely different process integration compared to the W integration process and the Al integration process, and is applied to the integration process of most logic device processes. Filling performance may vary slightly depending on the via patterning method, but there is no big difference in terms of filling characteristics. However, it is decided whether to apply W, Al, or Cu depending on the required device characteristics such as logic and memory. The above three materials can be applied depending on the process integration aspect and the material’s electrical resistance difference and electromigration characteristics. Currently, the Cu wiring process is applied to devices requiring high speed, and the W process is important in the interconnection area where junction contamination and high reliability are required.

Currently, it is common to develop high-speed logic devices using Cu wiring, but various logic devices that do not involve the requirement of high speed provide reasonable opportunities for optimizing the low-cost W/Al wiring process [21,22,23].

Ti deposition does not achieve sufficient bottom coverage by using the existing sputtering method, [24], and presently, the IMP method is the only applicable technique that shows the best characteristics, with the collimated PVD and long-throw PVD methods approaching their limits [25,26,27].

TiN deposition too has the drawback that sufficient bottom coverage cannot be obtained by using the existing sputtering method [28,29]. Only the metal organic CVD (MOCVD) process shows very good bottom-coverage characteristics [30,31]. After confirming the basic physical properties by verification, we examined the applicability of IMP Ti, CVD TiN, and CVD W to logic devices based on via resistance, and we established the optimal process conditions.

## 2. Experimental Procedure 

First, to confirm the applicability of the barrier film and W film to >150-nm logic devices, the physical and electrical characteristics of the via structure using IMP Ti, CVD TiN, and CVD W were examined. In addition, we attempted to find the most optimal process conditions by determining the resistance of the wiring structure in the logic device, based on the confirmed via resistance data. AMAT Endura was used to deposit the barrier film before the W deposition [27,32]. This equipment is composed of RF-etched (pre-clean) [33] IMP Ti, IMP TiN, and CVD TiN chambers, as shown in Figure 1. It provides a continuous process of RF etching–IMP Ti–CVD TiN (or IMP TiN) without air break.

The resistivity of each film is shown in Table 1, and the deposition conditions are shown in Table 2. The resistivity was obtained by depositing 5000 Å of plasma-enhanced tetraethylorthosilicate (PETEOS) on the bare wafer, 200 Å of IMP Ti [34,35], 100 Å of CVD TiN, and 300 Å of IMP TiN [34], and measuring the sheet resistance using an Omnimap [36,37,38,39].

CVD of W was performed using a Concept-1 Novellus instrument. The equipment had five heat stations and shower heads, each of which could be manipulated in a large chamber to control the temperature according to the deposition step. Another feature was the use of a minimum-overlap exclusion ring (MOER) to prevent W from being deposited on the wafer edge. Thus, the possibility of W residue remaining on the wafer edge after the W CMP process was fundamentally excluded. Deposition conditions at this time were SiH_4_ 0.025 slm, H_2_ 6 slm, and WF 6 0.28 slm [34,39,40]. The temperature was 395 °C, and the SiH_4_ reduction time of the W seed formation step was 12 s [41,42]. The step coverage and filling of the thin films of CVD TiN, IMP TiN, and CVD W were analyzed using SEM and TEM, and the etching profiles were analyzed using SEM after sample treatment using precision etching coating system (PECS; Gatan, Pleasanton, CA, USA) equipment.

## 3. Results and Discussion

### 3.1. Physical Characteristics of Via Profiles, Step Coverages of Thin Film, and Via Filling

As shown in Table 1, the uniformity of the IMP process is high, and the resistivity of CVD TiN is relatively high. The amount of CVD TiN applied was 2 × 50 Å and IMP TiN was used without the AC bias. In addition to the application of CVD TiN to the 150-nm-class logic device, a comparative evaluation was performed to analyze the IMP TiN process in terms of process optimization.

First, each film was deposited using a wafer with well-defined contact holes to confirm the bottom coverage of IMP Ti and CVD TiN [43]. Based on the 150–180-nm design rule for logic devices, the focus of the masking/etching process was also on the contact define, so we measured the hole profile at 0.24 hole size. Of course, there will be effects such as hole profile or pre-cleaning, but the last thing is the bottom-coverage ability at the largest aspect ratio like the contact hole, so if the bottom coverage at the contact is checked, the application in the via is also problematic. Deposition was performed by sequentially depositing CVD TiN 100 Å after IMP Ti 150 Å and excluding the RF-etch cleaning step. This wafer was analyzed using cross-sectional TEM to examine the thickness of the film deposited on the bottom of the hole, and the resulting images are shown in Figure 2. As depicted in the figure, the hole depth was approximately 7700 Å, the bottom CD was 0.24 µm, and the A/R was approximately 3.2:1. Under these conditions, the IMP Ti deposited on the floor was measured to be approximately 85 kPa, and the CVD TiN was approximately 60 kPa. The surface deposition thicknesses were 150 Å and 100 Å, respectively. Based on these results, the bottom coverage for the deposition conditions in Table 2 was found to be approximately 57% for IMP Ti and approximately 80% for CVD TiN at an A/R of 3.2:1; the via D/R of the 015 logic device was 0.22 (IMD 8000 (A/R 3.6:1)).

The CVD W film was deposited to evaluate the resistivity. In the same manner as followed for the resistivity measurement of the barrier film under the conditions discussed in Section 2, the sheet resistance was measured using an Omnimap after deposition of 3800 Å of W, and the thickness was measured using cross-sectional SEM to determine the resistivity of W. The calculated specific resistance was approximately 11.5 μΩ∙cm. Next, to confirm the via filling state under the current deposition conditions, the W filling state was confirmed using a 0.22-µm via hole size. Figure 3 shows the cross-sectional SEM images of via 1 and via 2 after sample processing using PECS. As shown in the figure, via 1, which had a poor via etch profile, was not filled with W, and a void existed at the center; via 2, which had a better profile, was well filled with W.

### 3.2. Electrical Characteristics of Via Structure

To optimize the MLM setup, the via resistance was measured using a 0.22-size via on a patterned wafer, to analyze the electrical properties as well as the basic film properties. The experimental conditions were measured in via 1, via 4, and via 2, and the details of each experimental condition are shown in Table 3. The effects of RF etching and IMP Ti deposition were investigated on via 1, and the CVD TiN and CVD W deposition conditions were tested on via 4; finally, the results were applied to via 2.

The resistance of via 1 according to the amount of RF etching applied to it is shown in Figure 4. At this time, the RF-etching conditions were an RF power of 400 W, RF secondary power of 275 W, and AR gas of 10 sccm. The RF etching split is a value measured based on the thermal oxide plate by adjusting the etching target according to changes in the etching time. The measurement pattern is a chain pattern of 300 via holes. In this case, the wire width of metals 1 and 2 is 0.8 µm, which is the size of the via resistance measurement pattern. As shown in Figure 4, most of the module parts defined as 0.16 µm failed, and the via resistance in the module defined as 0.21 µm decreased with continued RF etching. It is confirmed that a resistance value of less than 10 kV can be obtained when it is more than 200 kV. However, in the bottom misalignment pattern, the resistance of the via increases as the misalignment increases, but the via resistance of each RF-etching split shows a different trend from the via pattern with the minimum width. In other words, the resistance value was less than approximately 10 Ω without significant change, depending on the etching target, and the highest via resistance was observed when the amount of RF etching was 150 Å. From the RF-etching target split of via 1, we can conclude that the etching target should be more than 200 Å.

The via resistance according to IMP Ti thickness is shown in Figure 5. In the case of the minimum via size pattern, failure occurred in the test pattern defined as 0.16 µm, and the via resistance decreased as the IMP Ti deposition thickness increased in the 0.21 and 0.25 µm patterns.

In the 0.21-µm pattern, the results of the 100 Å and 150 Å conditions were the worst, whereas those of 250 Å and 300 Å were the best. In the case of the bottom-misaligned via pattern, the resistance distribution showed no difference according to the misalignment in the 0.00 and 0.04 misalignments, as in the RF-etching target split. The highest via resistance was identified. Based on this, it is considered that the IMP Ti thickness of 150 Å is insufficient, and a value of at least 200 Å is required.

The CVD TiN process is performed by repeating the deposition and plasma treatment using tetrakis dimethylamino titanium (TDMAT) as the precursor, and the plasma treatment effect is known to be the best at 50 kHz [44]. CVD TiN is a subsequent process, which does not contribute directly to the via resistance [45]. Because it acts as a barrier to prevent the penetration of F ions in the W deposition process, 1 × 50 Å and 3 × 50 Å were applied based on the results for 2 × 50 Å. At this time, both RF etching and the IMP Ti target were fixed at 150 Å, and the result is shown in Figure 6.

In the figure, 3 × 50 Å showed the lowest resistance and a stable tendency for pattern sizes of 0.22 µm and 0.26 µm, whereas 1 × 50 Å showed the poorest result. In addition, the resistance of 2 × 50 Å varied by wafer in the 0.22-µm via structure, but it can be seen that the value is less than approximately 10 Ω. Based on these results, CVD TiN as a via barrier in the via structure of CMOS logic interconnection is expected to show satisfactory performance in the 2 × 50 Å process, whereas the 3 × 50 Å process yields very good results.

Next, IMP TiN was applied as the via barrier, which has the advantages of lower resistivity, process time, and cost than CVD TiN [42,46]. IMP TiN was mounted with CVD TiN, as shown in Figure 1. The resistivity and deposition conditions of the film are shown in Table 1; Table 2, respectively. The deposition thickness of IMP TiN was similar to that of CVD TiN in as much as the amount deposited on the bottom of the via hole was equal to CVD TiN. Based on 80% and 57% bottom coverage of CVD TiN and IMP Ti, respectively, calculated at the 3.2:1 A/R, the bottom coverage of IMP Ti corresponded to a 3.5:1 A/R. When the coverage was 51%, the deposition thickness of IMP TiN was determined to be 200 μs based on the condition that the coverage of IMP TiN was 38%. This result is shown in Figure 7a. IMP TiN showed good results based on the results of the via resistance alone. This may be due to its lower resistivity than CVD TiN, but it should be considered that RF etching and IMP Ti are not optimized to 0.22-µm size, and addition of the IMP process can cause device damage. That is, as discussed above, the via resistance is not significantly degraded by the current CVD TiN condition, and as shown in Figure 7c, the via resistance is significantly improved when the RF etching and IMP Ti conditions are improved. It would be reasonable to use CVD TiN and consider the applicability of IMP TiN rather than select IMP TiN by a simple comparison with IMP TiN. The applicability of IMP TiN as a via barrier is considered satisfactory.

Figure 7b shows the results of the via resistance when W was deposited under different deposition conditions. The process temperature of 350 °C corresponds to an A/R of approximately 9:1, and the respective deposition conditions are shown in Table 4.

The difference between the results of the 256LD and 015 logic devices occurs because the deposition temperature (350 °C) of 256LD is significantly lower, and the nucleation time is short. The nucleation thickness was approximately 400 µs under the deposition conditions for the 015 logic device, and the nucleation under the conditions applied for 256LD is expected to be thinner due to the short nucleation time. The results for the via resistance show that the variation between the wafers is severe, and the difference in each condition is within the wafer deviation, making it difficult to establish an accurate comparison; however, there is no difference between the two conditions. In other words, the effect of nucleation thickness and deposition temperature on the via resistance does not appear to be significant. Based on the split results in Figure 7c, we compared the via resistance values between the conditions expected to be the most suitable for the 015 logic via barrier and the other existing conditions. The optimal conditions were selected as RF etching 200 [47], IMP Ti 200, and CVD TiN 2 × 50 Å. The results show that the obtained via resistance is less than approximately 5 kW, and the distribution of resistance is very good.

## 4. Conclusions 

The vias of logic devices can be filled by applying W plugs, and the RF etching 200 Å IMP Ti 200 Å CVD TiN 2 × 50 Å barrier structure provides stable and excellent via resistance. The conditions for optimization were selected.

The possibility of using IMP TiN as a barrier in the via was confirmed. The resistivities of IMP Ti and CVD TiN were 73 and 280 μΩcm, and the bottom coverages were 57% and 80%, respectively, at a 3.2:1 A/R. The specific resistance of the W film was approximately 11.5 μΩcm, and it was confirmed that the via filling could be performed smoothly. The RF etching and IMP Ti should be at least 200 Å. W deposition showed no difference in the via resistance with deposition temperature and SiH_4_ reduction time. Therefore, RF etching 200 Å IMP Ti 200 Å CVD TiN 2 × 50 Å yielded the best results.

## Figures and Tables

**Figure 1 micromachines-11-00032-f001:**
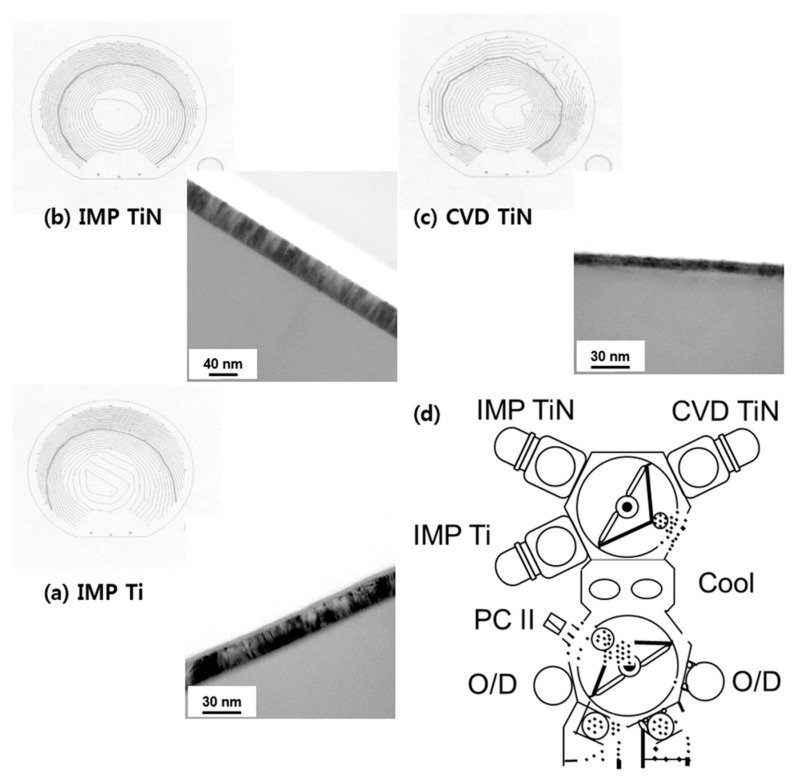
TEM image and Omnimap of (**a**) IMP Ti 200 Å, (**b**) IMP TiN 300 Å, and (**c**) CVD TiN 100 Å (2 × 50 Å). (**d**) Schematic diagram of cluster tool containing IMP TiN deposition chamber, IMP Ti deposition chamber, pre-clean chamber (PC II), cooling chamber, and CVD TiN deposition chamber.

**Figure 2 micromachines-11-00032-f002:**
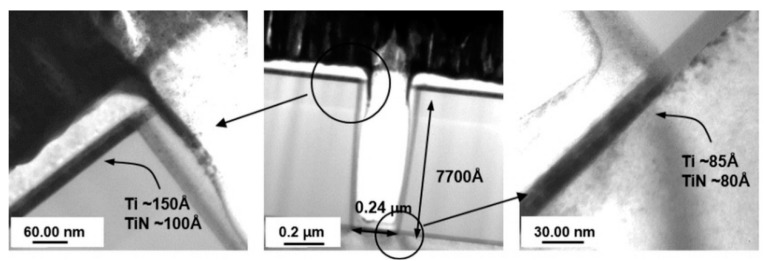
TEM image of cross section after each film is deposited using HRS0005 with complete contact hole definition.

**Figure 3 micromachines-11-00032-f003:**
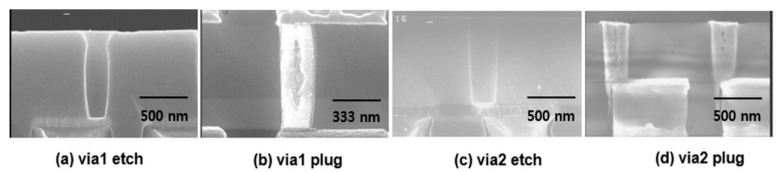
The cross section SEM images of after (**a**) via 1 etch, (**b**) via1 plug W filling, (**c**) via2 etch, and (**d**) via 2 plug W filling.

**Figure 4 micromachines-11-00032-f004:**
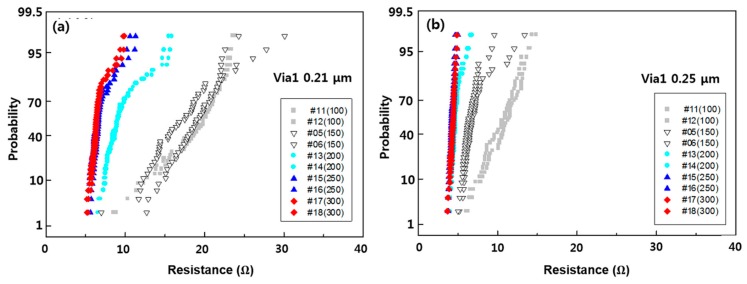
Via resistance according to IMP Ti thickness in minimum via size pattern (**a**) via 1 0.21 μm, (**b**) via 1 0.25 μm.

**Figure 5 micromachines-11-00032-f005:**
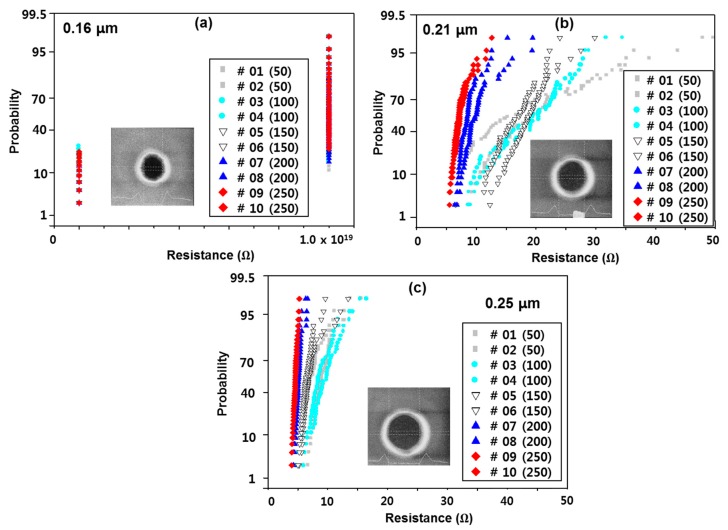
etch critical demension (FICD) SEM image of (**a**) 0.16, (**b**) 0.21 and (**c**) 0.25 via 1 insistence (minimum width via pattern).

**Figure 6 micromachines-11-00032-f006:**
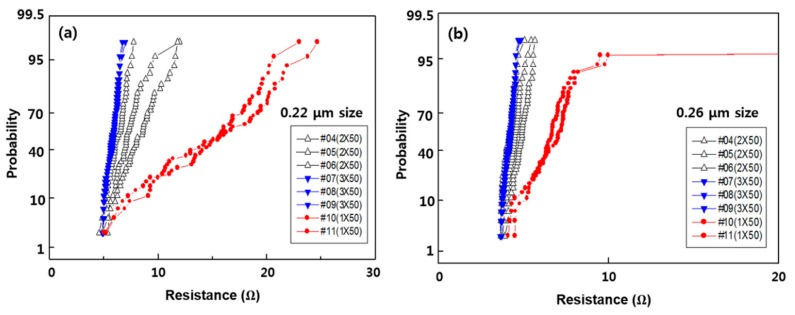
Via resistance according to CVD TiN thickness of (**a**) 0.22 μm and (**b**) 0.25 μm in minimum via size pattern.

**Figure 7 micromachines-11-00032-f007:**
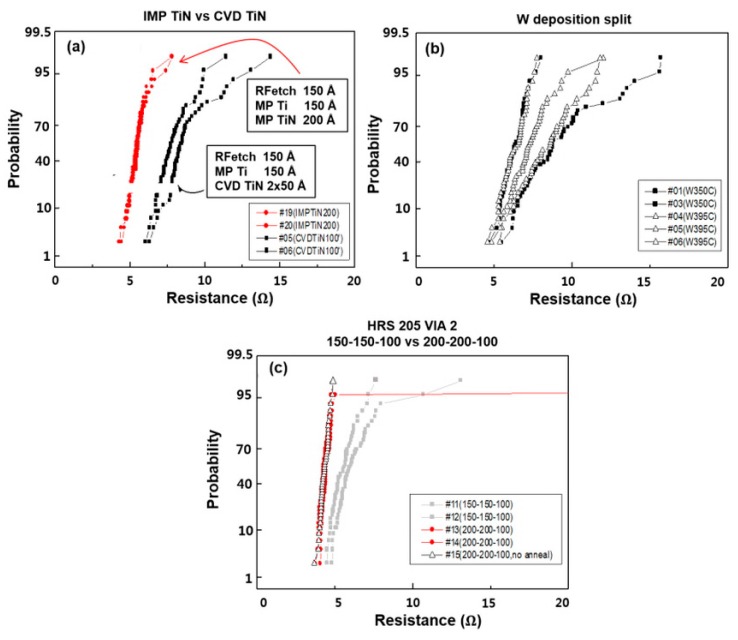
(**a**) Comparison of via resistance between IMP TiN and CVD TiN. (**b**) Comparison of results of change of W deposition temperature 350 °C. (**c**) 015 logic via barrier evaluation of via resistance in deposition conditions.

**Table 1 micromachines-11-00032-t001:** Sheet resistance (*R*s) and specific resistance value (*ρ*) for each film.

Analysis Item	IMP Ti	CVD TiN	IMP TiN
Average *R*s (Ω/sq)	39.9	302.05	29.06
Unif. (%)	4.99	3.55	9.05
TEM phase thickness (center, Å)	195	100	345
*R*s (center, Ω/sq)	37.5	280.8	24.73
Resistivity (μΩ∙cm)	~73	~280	~85
Stress (dyne∙cm^2^)	−2.142 × 10^−9^	−4.788 × 10^−9^	−6.762 × 10^−9^

**Table 2 micromachines-11-00032-t002:** Deposition conditions for each film.

**IMP Ti**	DC Power		RF Power	AC Bias		Ar	
	2250 W		2750 W	0 W		56 sccm	
**CVD TiN**	Deposition Condition			PLASMA TREAT			
	Pressure	TEMP	He carr	Pressure	RF power		TEMP
	1.5 Torr	450 °C	225 sccm	1.3 Torr	750 W		450 °C
IMP TiN	DC power	RF power	AC bias	Ar	N_2_		
	4000 W	2500 W	0 W	25 sccm	28 sccm		

**Table 3 micromachines-11-00032-t003:** Split item for each step.

Step	Wafer No.	RF Etch Target	IMP Ti Target	CVD TiN Target
Via 1	01, 02	50	150	2 × 50
	03, 04	100		
	05, 06	150		
	07, 08	200		
	09, 10	250		
	11, 12	150	100	
	13, 14		200	
	15, 16		250	
	17, 18		300	
	19, 20		150	IMP TiN 200
Via 4	01, 02, 03	150	150	2 × 50
	04, 05, 06	150	150	2 × 50
	07, 08, 09	150	150	3 × 50
	10, 11, 12	150	150	1 × 50
Via 2	11, 12	150	150	2 × 50
	13, 14, 15	200	200	2 × 50

**Table 4 micromachines-11-00032-t004:** Tungsten (W) deposition conditions.

Item		015 Logic	256 LD
**Gas**	SiH_4_ (slm)	0.025	0.025
	H_2_ (slm)	6	6
	WF_6_ (slm)	0.28	0.28
Temp.	T1 (°C)	395	395
	T2 (°C)	395	350
	T3 (°C)	395	350
	T4 (°C)	395	445
	T5 (°C)	475	475
Nucleation time	SiH_4_ reduction (sec)	12	8
Deposition thickness (Å)		3800	4000

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
