# Peer review of "Process Optimization of Via Plug Multilevel Interconnections in CMOS Logic Devices"

_micromachines, 2019, doi:10.3390/mi11010032_

Round 1

Reviewer 1 Report

The manuscript by Cui et al. entitled “Process optimization of via plug multilevel interconnection in CMOS logic device” reports the optimization of depositing multilevel interconnection for via plug in CMOS logic devices. Even though the authors compared different deposition conditions for the interconnection, detailed scientific explanation why RF etching 200, IMP Ti 200, and CVD TiN 2X50 shows optimal results is not shown in the manuscript. Comparison of the results in the manuscript with literature is also required. Figures need further improvement. For example, inset images in Figure 1 lack description. Drawing of equipment in Figure 1 should be labeled as Figure 1(d) and better drawing is expected. Writing needs significant improvement. Other mistakes/typos like “The surface deposition thicknesses were 150 kPa and 100 kPa, respectively.” on page 4 should be corrected. Due to abovementioned issues, I am fairly inclined to reject the manuscript to be published on Micromachines.

Reviewer 2 Report

The paper talks about some optimization for via plug multilevel interconnections. There are some issues to be addressed before proceeding to be accepted. 

Can the authors add more detailed information in the introduction part to clarify some claims? a. The Al filling limit for the Vias; b. The applicability of W compared to the widely used Al filling and Cu Damascene; c. The use of Ti. With all the above information, they need to reorganize the background and introduction to have more clear flow and convincing motivation. 

Can the authors compare the properties, for example, the resistance, of their samples with previously reported ones using similar processes? 

In Figure 2, it seems the some space was not filled well at the opening region, can the authors explain more? In figure 3, Via 1 etching profile looks not so good as via 2 etch?  From figure 4, it seems to me the process can not well applied to very fine features smaller than 210nm? 

Can the authors talk more about the limitations for their process, and the effect on the capacitance between the interconnects?

Author Response

Point 1: The paper talks about some optimization for via plug multilevel interconnections. There are some issues to be addressed before proceeding to be accepted. 

Can the authors add more detailed information in the introduction part to clarify some claims? a. The Al filling limit for the Vias; b. The applicability of W compared to the widely used Al filling and Cu Damascene; c. The use of Ti. With all the above information, they need to reorganize the background and introduction to have more clear flow and convincing motivation. 

Can the authors compare the properties, for example, the resistance, of their samples with previously reported ones using similar processes? 

Response 1: Please provide your response for Point 1. (in red)

We thank you for considering our submission and for your valuable comments. The manuscript has been carefully revised according to the reviewer’s comments. We have addressed the reviewer’s comments individually below.

Thank you again for the reviewer's comment.

As described in the reviewer's comment, the introductory description is as follows and references are also inserted.

Via filling processes include W plug fill process, Al plug fill process and Cu damascene fill process. In the W plug fill process, the CVD W process is applied in terms of filling. [15, 16] It is essential to derive the optimum conditions of via profile photo/etch integration process, glue layer Ti, and barrier metal TiN process to optimize the process. If the Ti / TiN process is not optimized, the Volcano effect [17, 18] can occur, causing a fatal defect that can cause the device to fail. In the Al plug fill process, a two step process of CVD Al / PVD Al is applied, and the selection of precusor, flow fill method, glue layer and pattern profile applied to CVD Al act as process variables. Currently, the pattern of sub-0.2mm (aspect ratio) 4 vias is a voidless and stable filling process with an integrated CVD Al / PVD AlCu approach process [19, 20]. Cu damascene process is a completely different process integration process compared to W integration process and Al integration process, and is applied to process integration process of most logic device process. Filling performance may vary slightly depending on the via patterning method, but there is no big difference in terms of filling characteristics. However, it is decided whether to apply W, Al, or Cu depending on the required device characteristics such as logic and memory. The above three materials can be applied depending on the process integration aspect and the material's electrical resistance difference and electromigration characteristics. Currently, the Cu wiring process is applied to devices requiring high speed, and the W process is important in the interconnection area where junction contamination and high reliability are required.

Reference

Ou, N.C.; Bock, D.C.; Su, X.; Craciun, D.; Craciun, V.; McElwee-White, L. Growth of WO x from Tungsten (VI) Oxo-Fluoroalkoxide Complexes with Partially Fluorinated β-Diketonate/β-Ketoesterate Ligands: Comparison of Chemical Vapor Deposition to Aerosol-Assisted CVD. ACS Appl. Mater. Interfaces 2019, 11, 28180–28188. Lee, P.; Cronin, J.; Kaanta, C. Chemical vapor deposition of tungsten (CVD W) as submicron interconnection and via stud. J. Electrochem. Soc. 1989, 136, 2108–2112. Chang, H.L.; Juang, F.L.; Kuo, C.T. Effect of silane flowing time on W volcano and plug formation. Jpn. J. Appl. Phys. 2002, 41, 2906. Kraft, J.; Stückler, E.; Cassidy, C.; Niko, W.; Schrank, F.; Wachmann, E.; Gspan, C.; Hofer, F. Volcano effect in open through silicon via (TSV) technology. In Proceedings of the 2012 IEEE International Reliability Physics Symposium (IRPS); IEEE, 2012; p. PI-2. El-Kareh, B.; Hutter, L.N. Process Integration. In Silicon Analog Components; Springer, 2020; pp. 447–494. Dixit, G.A.; Paranjpe, A.; Hong, Q.-Z.; Ting, L.M.; Luttmer, J.D.; Havemann, R.H.; Paul, D.; Morrison, A.; Littau, K.; Eizenberg, M. A novel 0.25/spl mu/m via plug process using low temperature CVD Al/TiN. In Proceedings of the Proceedings of International Electron Devices Meeting; IEEE, 1995; pp. 1001–1004.

Point 2: In Figure 2, it seems the some space was not filled well at the opening region, can the authors explain more? In figure 3, Via 1 etching profile looks not so good as via 2 etch?  From figure 4, it seems to me the process can not well applied to very fine features smaller than 210nm? 

Response 2: Thank you for the reviewer's comment.

Fig. 2 shows the step coverage characteristics of the etch profile and glue layer and barrier metal in the pre-w filling process step and is not related to the filling characteristics. As shown in Figure 3, via 1 etch profile shows different via2 etch profile due to integration characteristics. Figure 4 shows the FICD, which can be optimized according to the process conditions.

Point 3: Can the authors talk more about the limitations for their process, and the effect on the capacitance between the interconnects?

Response 3: Thank you again for the reviewer's comment.

The current optimized process can be applied to the ~ 0.18um via process. The capacitance between interconnects is not measured in the present experiment, but the capacitance is mainly determined by the dielectric constant of the inter-metal dielectric (IMD). The RF etch process causes dielectric loss due to the metal deposition process, which may lead to cross talk and capacitance changes.

Reviewer 3 Report

The work presented in this manuscript is useful for the scientific community working in this field. All conclusions are correct and supported by the experiment, analysis and discussion. I recommend to publish this manuscript after a short improvements.

Figure 4 is too small compared to the rest of the figures. The values ​​are not visible. In fact all the figures in the article have different sizes. I think that all should be brought to a common dimension. Please check the representation in Figure 7. A graph appears to be colorless.

What structure have the deposited layers (amorphous or crystalline)? Nowhere does this look. It would also help AFM images for a complete characterization of the surfaces. Can they predict what results they would have achieved if the layers they obtained were crystalline or amorphous?

It was annealed films before measurements? Was there any surface treatment step before electrical measurements? The performance of the device structure can be significantly affected by semiconductor film thickness in time of temperature measurements.

The electrical measurements are made on top structures or bottom structures? Did you make both measurements? Did you observe differences in this kind of measurements?

Hope my suggestions are helpful.

Author Response

Point 1: The work presented in this manuscript is useful for the scientific community working in this field. All conclusions are correct and supported by the experiment, analysis and discussion. I recommend to publish this manuscript after a short improvements.

Figure 4 is too small compared to the rest of the figures. The values ​​are not visible. In fact all the figures in the article have different sizes. I think that all should be brought to a common dimension. Please check the representation in Figure 7. A graph appears to be colorless.

Response 1: Please provide your response for Point 1. (in red)

We thank you for considering our submission and for your valuable comments. The manuscript has been carefully revised according to the reviewer’s comments. We have addressed the reviewer’s comments individually below.

Thank you again for the reviewer's comment.

Fig. 4 revises the picture according to the reviewer's comment.

Point 2: What structure have the deposited layers (amorphous or crystalline)? Nowhere does this look. It would also help AFM images for a complete characterization of the surfaces. Can they predict what results they would have achieved if the layers they obtained were crystalline or amorphous?

Response 2: Thank you for the reviewer's comment.

The deposited metal films are in crystalline phase. Surface characteristics are removed in the subsequent integration process, CMP process, so AFM analysis is not performed separately.

Point 3: It was annealed films before measurements? Was there any surface treatment step before electrical measurements? The performance of the device structure can be significantly affected by semiconductor film thickness in time of temperature measurements.

Response 3: Thank you again for the reviewer's comment. Electrical measurements were carried out to the probe station in semiconductor line, and there was no surface treatment because the probe was going into the membrane to some extent. In addition, the W film has almost no change in characteristics with time, and it is judged that there is no effect of time delay since the process proceeds to continuous measurement in the semiconductor line after the process proceeds.

Point 4: The electrical measurements are made on top structures or bottom structures? Did you make both measurements? Did you observe differences in this kind of measurements?

Response 4: Thank you again for the reviewer's comment..

Electrical measurement measurement method is a test method of semiconductor company and it is judged that there is no difference.

Round 2

Reviewer 1 Report

The authors' reply answered most of my questions. But following shortcomings needs to be corrected. On line 62, the authors cited Ref.10 to show Ti deposition doesn't have enough bottom coverage using existing sputtering method. However, Ref. 10 only talks about W deposition and is irrelevant to Ti deposition. On line 67, the authors cited Ref. 30 to show MOCVD provides better bottom coverage. But, in Ref. 30, MOCVD grown SiN is in situ deposited on GaN epiwafer after growth of GaN epilayer and the epiwafer doesn't have via holes at all.

The cited works in this manuscript needs careful examination to avoid any misleading information for readers in the future.

Reviewer 2 Report

The authors addressed my concerns, and the paper is good to go.